# Vitamin A supplements, routine immunization, and the subsequent risk of *Plasmodium* infection among children under 5 years in sub-Saharan Africa

Maria-Graciela Hollm-Delgado[1]*, Frédéric B Piel[2], Daniel J Weiss[3], Rosalind E Howes[3], Elizabeth A Stuart[4], Simon I Hay[3], Robert E Black[1]

[1]Department of International Health, Bloomberg School of Public Health, Johns Hopkins University, Baltimore, United States; [2]Evolutionary Ecology of Infectious Disease Group, Department of Zoology, University of Oxford, Oxford, United Kingdom; [3]Spatial Ecology and Epidemiology Group, Department of Zoology, University of Oxford, Oxford, United Kingdom; [4]Departments of Mental Health and Biostatistics, Bloomberg School of Public Health, Johns Hopkins University, Baltimore, United States

**Abstract** Recent studies, partly based on murine models, suggest childhood immunization and vitamin A supplements may confer protection against malaria infection, although strong evidence to support these theories in humans has so far been lacking. We analyzed national survey data from children aged 6–59 months in four sub-Saharan African countries over an 18-month time period, to determine the risk of *Plasmodium* spp. parasitemia (n=8390) and *Plasmodium falciparum* HRP-2 (*Pf*HRP-2)-related antigenemia (n=6121) following vitamin A supplementation and standard vaccination. Bacille Calmette Guerin-vaccinated children were more likely to be *Pf*HRP-2 positive (relative risk [RR]=4.06, 95% confidence interval [CI]=2.00–8.28). No association was identified with parasitemia. Measles and polio vaccination were not associated with malaria. Children receiving vitamin A were less likely to present with parasitemia (RR=0.46, 95% CI=0.39–0.54) and antigenemia (RR=0.23, 95% CI=0.17–0.29). Future studies focusing on climate seasonality, placental malaria and HIV are needed to characterize better the association between vitamin A and malaria infection in different settings.

**\*For correspondence:** mhollmd1@jhu.edu

**Reviewing editor**: Stephen Tollman, Wits University, South Africa

## Introduction

Malaria remains a major public health challenge, with more than 7% of deaths among children under 5 years in the world attributable to the disease (*Liu et al., 2012*). Many of these cases arise in regions with high universal immunization coverage. In sub-Saharan Africa alone where 80% of malaria cases occur (*World Health Organization, 2012*), district coverage rates are estimated at over 70% for the third dose of diphtheria–tetanus–pertussis (DTP) vaccines and 75% for measles-containing vaccines (*World Health Organization and UNICEF, 2013*).

Several studies have suggested that childhood immunization may confer non-specific health effects beyond targeted diseases, but the impact of standard vaccination and vitamin A supplementation on malaria infection is unclear. In particular, early evidence from murine models indicates that injection with live strains of Bacille Calmette Guerin (BCG) vaccine may confer sterilizing protection against *Plasmodium* infection (*Clark et al., 1976*). However, this effect seems to depend on several factors including route of administration (intradermal, subcutaneous, intramuscular vs intravenous), type of immunogen used (killed vs live attenuated), vaccine schedule in relation to malaria exposure

**eLife digest** More than half of the world's population is at risk of malaria, with an estimated 198 million clinical cases each year. A vaccine that fully prevents it has not yet been discovered. Most cases of malaria occur among children living in sub-Saharan Africa, a region where many receive routine vaccinations designed to prevent other diseases; for example, 75% of children in sub-Saharan Africa receive measles vaccines. Many also receive vitamin A supplements, which have been linked not only to the protection of a child's vision, but also to a lower risk of death and an improved ability to fight off infections.

Some researchers have suggested that vitamin A supplements and routine childhood vaccinations for other diseases may also provide some protection against malaria. For example, some studies performed in mice have shown that a commonly used tuberculosis vaccine may eliminate *Plasmodium* parasites that cause malaria infections. However, this effect depended on several factors, including how the vaccine was administered and whether the vaccination was given before or after the mouse developed malaria.

It is less clear whether vaccines or vitamin A have antimalarial effects in humans. To address this, Hollm-Delgado et al. analyzed national survey data collected from thousands of children aged between 6 months and 5 years old who lived in four different countries in sub-Saharan Africa. The surveys contained information about the vaccines and supplements the children received, and whether their blood showed signs of infection with malaria-causing *Plasmodium* parasites.

Hollm-Delgado et al. found that routine vaccinations did not affect the likelihood of malaria parasites being detected in the child's blood. However, children vaccinated against tuberculosis were more likely to have a specific type of protein released when malaria infects the blood. Hollm-Delgado et al. suspect that the tests may actually have inadvertently detected other parasitic infections in the children, such as *Schistosoma*, producing false-positive results for malaria.

In contrast, Hollm-Delgado et al. found that children who received vitamin A supplements were less likely to become infected with malaria. The benefits of the supplements appeared to be affected by several conditions, including the time of year when the children received their supplements or when they were tested for malaria, and whether their mother had malaria when pregnant. Clinical trials are now needed to confirm these results and investigate how effectively vitamin A prevents malaria.

(before vs after infection), mouse strain, sex, and *Plasmodium* species tested (*Clark et al., 1976*; *Smrkovski and Strickland, 1978*, *Smrkovski, 1981*, *1982*; *Murphy, 1981*, *Matsumoto et al., 2000*). Conversely, the risk of malaria infection has been thought to increase during downregulation in cellular immunity (including interleukin-12, interferon-gamma, T cells) and T helper (Th) 2 cell upregulation following measles infection and/or immunization (*Desai et al., 2005*). Epidemiological evidence to support this theory in humans remains unclear (*Whittle et al., 1980*; *Rooth and Bjorkman, 1992*).

Multiple clinical trials in infants and young children have also identified protective effects from vitamin A supplementation against illness (number and time to first clinical episode, risk of febrile illness, spleen enlargement, and mean parasite density) (*Shankar et al., 1999*; *Zeba et al., 2008*) and death caused by malaria (*Fawzi et al., 1999*; *Varandas et al., 2001*; *Mwanga-Amumpaire et al., 2012*). Vitamin A is thought to act as a regulator of pro-inflammatory response genes (e.g. tumor necrosis factor alpha) and phagocytotic clearance (e.g. cluster of differentiation 36 [CD36]) of *Plasmodium falciparum*-infected erythrocytes, by way of its active metabolite, retinoic acid (*SanJoaquin and Molyneux, 2009*). Less certain is its impact on malaria infection. While vitamin A did not appear to modulate the risk of parasitemia in a clinical trial study of Ghanaian children (*Binka et al., 1995*) and Tanzanian children previously hospitalized with pneumonia (*Villamor et al., 2003*), a cluster randomized intervention trial for breastfeeding in Uganda observed a sixfold decrease in the adjusted risk of malaria infection among children receiving vitamin A (*Nankabirwa et al., 2011*). On the other hand, concern has been raised by an apparent increase in levels of parasitemia within mouse models, when combining vitamin A supplements with DTP vaccination; an effect thought to be stronger among female mice (*Jorgensen et al., 2011*). The underlying mechanisms of these effects are unclear,

although it has been theorized that vitamin A may act as adjuvant to DTP, which is postulated to cause detrimental effects (*Jorgensen et al., 2011*).

Based on this evidence, the purpose of this study was to determine the risk of *Plasmodium* parasitemia and *P. falciparum*-specific antigenemia following vitamin A supplementation and standard vaccination in children under 5 years of age in sub-Saharan Africa.

## Results

### Children's characteristics

Of 20,984 children who presented health cards during survey interviews, 18,413 were eligible for blood screening from which 12,058 provided capillary blood for malaria testing (*Figure 1*). From these, 8672 (72%) were tested using both thick blood films and rapid *P. falciparum* histidine rich protein-2 (*Pf*HRP-2) tests, 3356 (28%) were tested only with blood films, and 30 (0.2%) were tested with only rapid *Pf*HRP-2 tests. Among those tested, we identified 3544 (30%) with positive blood films for *Plasmodium* spp. and 3131 (36%) with positive *Pf*HRP-2 antigenemia. Results from both tests were 87% concordant. Complete confounder information was available for 8390 (70%) subjects who were tested for parasitemia and 6121 (70%) tested for antigenemia. *Table 1* shows BCG, DTP, and poliomyelitis vaccines were used most often. Vitamin A was least used. *Supplementary file 3* shows that immunization schedules were similar across all countries (*World Health Organization and UNICEF, 2007*). Malaria was most common among subjects from Burkina Faso and least common among those from Rwanda. Although Rwanda had the longest rainy season, they also had the greatest ownership of bed nets, and were least likely to have recently used antimalarials or had a mother who used antimalarials during pregnancy. Although HIV testing was not conducted among children, the mothers of 2540 subjects were tested for HIV. Twenty-four (0.94%) were seropositive, suggesting that the

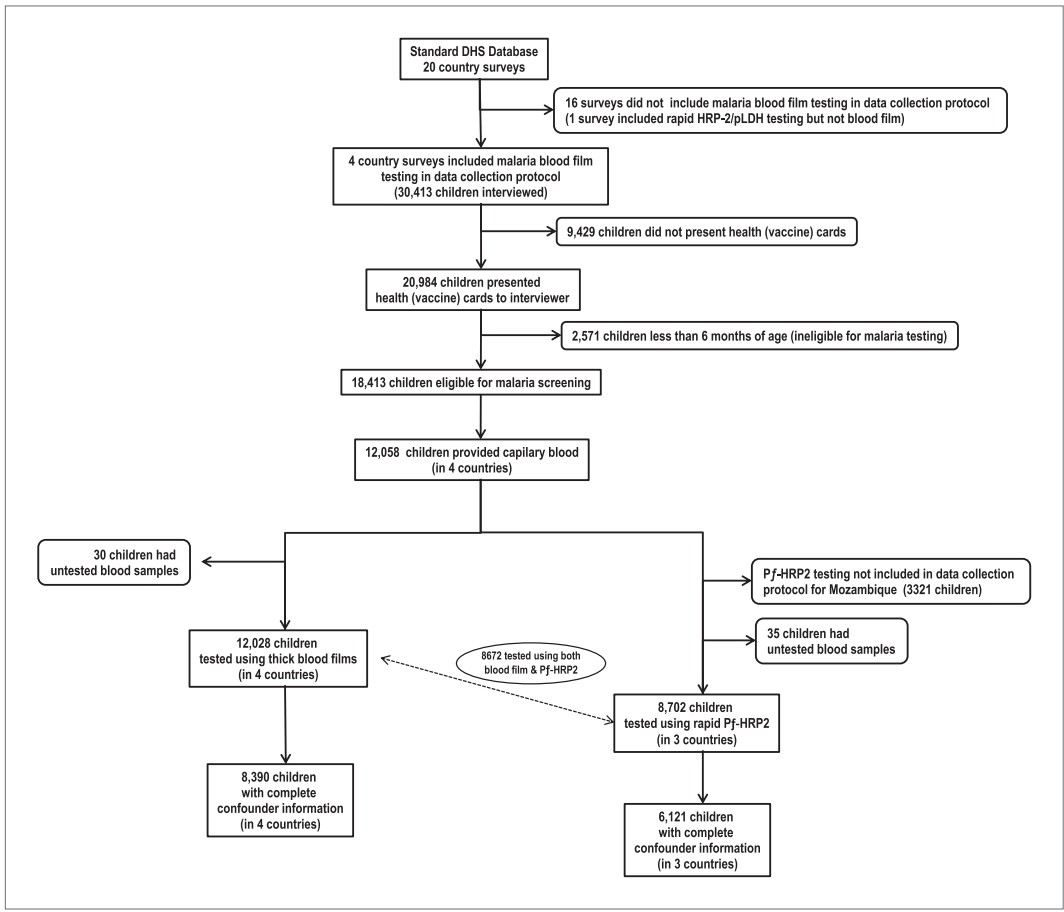

**Figure 1**. Flow chart of subjects.

**Table 1.** Baseline characteristics of 8390 children tested for malaria using blood film, by survey location

| Characteristic | Location of survey | | | | |
| | Burkina Faso (n=2821) | Mozambique (n=2266) | Rwanda (n=2085) | Senegal (n=1218) | Overall (n=8390) |
|---|---|---|---|---|---|
| Communities surveyed, n | 537 | 607 | 490 | 371 | 2005 |
| **(a) Children tested for malaria, n (%)** | | | | | |
| Parasitemia | 2821 (99) | 2266 (100) | 2085 (99) | 1218 (99) | 8390 (99) |
| Positive result among children tested for parasitemia | 1696 (60) | 578 (25) | 16 (0.8) | 22 (1.8) | 2312 (28) |
| Pf-HRP-2 | 2821 (100) | na | 2060 (99) | 1217 (99) | 6098 (99)* |
| Positive results among children tested for Pf-HRP-2 | 2051 (73) | na | 35 (1.7) | 27 (2.2) | 2113 (35) |
| **(b) Type of immunization received, n (%)** | | | | | |
| Bacille Calmette Guerin (BCG) | 2799 (99) | 2004 (96) | 2050 (100) | 1160 (98) | 8013 (98) |
| Diphtheria–tetanus–pertussis (DTP) | 2721 (97) | 2107 (97) | 2055 (98) | 1161 (98) | 8044 (98) |
| Measles | 2225 (80) | 1668 (78) | 1727 (86) | 830 (73) | 6450 (80) |
| Poliomyelitis | 2810 (100) | 2156 (99) | 2061 (100) | 1204 (100) | 8231 (99) |
| Vitamin A | 87 (9.3) | 1554 (87) | 351 (72) | 190 (59) | 2182 (62) |
| **(c) Children's characteristics** | | | | | |
| Age in months, median (IQR) | 22 (13–32) | 20 (13–31) | 24 (14–36) | 19 (12–29) | 22 (13–32) |
| Girls, n (%) | 1347 (48) | 1151 (51) | 1020 (49) | 551 (45) | 4069 (48) |
| Primigravidae, n (%) | 443 (16) | 477 (21) | 432 (21) | 251 (21) | 1603 (19) |
| Low birth weight, n (%) | 867 (31) | 1003 (44) | 672 (32) | 536 (44) | 3078 (37) |
| **(d) Malaria-based interventions, n (%)** | | | | | |
| Child's family owns bed net | 2160 (77) | 1528 (67) | 1983 (95) | 1040 (85) | 6711 (80) |
| Child received antimalarial during past week | 314 (11) | 117 (5.2) | 50 (2.4) | 24 (2.0) | 505 (6.0) |
| Child's house had indoor insecticide spraying | 22 (0.8) | 541 (24) | na | 148 (12) | 711 (8.5)† |
| Mother took antimalarial during child's gestational period | 2616 (93) | 1109 (50) | 337 (16) | 1110 (91) | 5162 (62) |
| **(e) Genetic mechanisms of malaria protection, median (IQR)** | | | | | |
| Mean predicted HbS allele frequency | 0.06 (0.05–0.06) | 0.03 (0.01–0.03) | 0.03 (0.03–0.03) | 0.07 (0.06–0.07) | 0.04 (0.03–0.06) |
| Median predicted G6PDd allele frequency | 0.06 (0.05–0.09) | 0.15 (0.15–0.17) | 0.04 (0.04–0.05) | 0.10 (0.09–0.13) | 0.08 (0.05–0.14) |
| **(f) Climate of communities surveyed, median (IQR)** | | | | | |
| Annual range of enhanced vegetation index (EVI) | 0.29 (0.24–0.32) | 0.33 (0.22–0.40) | 0.25 (0.20–0.29) | 0.28 (0.18–0.34) | 0.28 (0.22–0.33) |
| Annual mean of EVI for the year | 0.22 (0.18–0.25) | 0.40 (0.32–0.47) | 0.39 (0.36–0.42) | 0.18 (0.16–0.20) | 0.29 (0.20–0.40) |
| No. of days which EVI above annual mean | 136 (120–160) | 88 (56–128) | 320 (168–384) | 208 (184–264) | 152 (120–200) |
| No. of days for rainy season (corresponding to first and last day EVI above annual mean) | 136 (120–152) | 128 (72–344) | 344 (296–352) | 216 (152–352) | 168 (120–344) |

*Pf-HRP-2 testing not conducted as part of DHS survey protocol for Mozambique.
†Information on insecticide spraying not collected as part of survey.

potential for vertical transmission might be low. There were no significant differences in seropositivity across countries.

## Impact of routine immunization on malaria infection

*Table 2* shows that while BCG vaccination was not associated with *Plasmodium* parasitemia, it was linked to an increased risk of *Pf*HRP-2 antigenemia. This effect remained after controlling for patient characteristics associated with BCG use in the inverse probability weighted (IPW) model. DTP was associated with a lower risk of *Plasmodium* parasitemia and *Pf*HRP-2, but only in the weighted models. Measles and poliomyelitis vaccination were not associated with malaria antigenemia or parasitemia. *Figure 2A* shows that the strength of associations between BCG and antigenemia were greater if children were vaccinated during the wet season, among younger children and the more time passed since vaccination.

## Impact of vitamin A supplementation on malaria infection

Vitamin A supplementation appeared protective against *Plasmodium* parasitemia and *Pf*HRP-2 antigenemia (*Table 2*). These effects remained after weighting. *Figure 2* illustrates that the

**Table 2.** Relative risk of malaria infection after standard vaccination and vitamin A supplementation among children 6–59 months of age

| Type of immunization | No. of children vaccinated/total tested (%) | No. of children with positive blood test (%) | | Unadjusted RR | Adjusted RR (95% CI)‡,§ | |
|---|---|---|---|---|---|---|
| | | No vaccine | Vaccine | | Unweighted | Weighted (IPW) |
| (a) *Plasmodium* species (parasitemia)* | | | | | | |
| Bacille Calmette Guerin (BCG) | 8013/8140 (98) | 41 (32) | 2227 (28) | 0.81 | 1.25 (0.81–1.91) | 1.24 (0.76–2.05) |
| Diphtheria–tetanus–pertussis (DTP) | 8044/8235 (98) | 83 (44) | 2202 (27) | 0.49 | 0.88 (0.64–1.20) | 0.06 (0.01–0.47) |
| Measles | 6450/8069 (80) | 473 (29) | 1784 (28) | 0.93 | 1.11 (0.96–1.29) | 1.01 (0.20–5.19) |
| Poliomyelitis | 8231/8272 (99) | 14 (34) | 2278 (28) | 0.74 | 0.80 (0.37–1.73) | 0.74 (0.27–2.01) |
| Vitamin A supplement | 2182/3523 (62) | 596 (44) | 438 (20) | 0.31 | 0.46 (0.39–0.54) | 0.43 (0.36–0.52) |
| (b) *Plasmodium falciparum* (antigenemia) | | | | | | |
| Bacille Calmette Guerin (BCG) | 6006/6047 (99) | 9 (22) | 2102 (35) | 1.91 | 4.06 (2.00–8.28) | 3.52 (1.66–7.48) |
| Diphtheria–tetanus–pertussis (DTP) | 5933/6054 (98) | 59 (49) | 2049 (35) | 0.55 | 1.34 (0.88–2.02) | 0.06 (0.01–0.38) |
| Measles | 4776/5937 (80) | 410 (35) | 1679 (35) | 0.99 | 1.15 (0.97–1.38) | 0.68 (0.15–3.12) |
| Poliomyelitis | 6.072/6084 (99) | 5 (42) | 2111 (35) | 0.75 | 1.39 (0.55–3.49) | 0.93 (0.37–2.35) |
| Vitamin A supplement | 629/1749 (36) | 621 (56) | 75 (12) | 0.10 | 0.23 (0.17–0.29) | 0.22 (0.16–0.29) |

HRP-2: histidine rich protein-2; RR: relative risk; CI: confidence interval; IPW: inverse probability weighted model.

*Tested in four countries: Burkina Faso, Mozambique, Rwanda and Senegal.

†Tested in three countries: Burkina Faso, Rwanda and Senegal.

‡Adjusted for the following factors: age, gender, wealth index score, mother's highest level of education, malaria treatment during previous week, ownership of bed net, proportion of household members under 5 years using bed net during previous night, indoor household insecticide spraying, mother's access to antenatal care during last pregnancy, mother's knowledge regarding vertical HIV transmission, malaria transmission season, and type of community setting (urban vs rural).

§Inverse probability weighting (IPW) based on propensity score model with following factors: age, gender, low birth weight, presence of radio or television, urban versus rural setting, breastfeeding status, wealth index score, mother's age, mother's highest education level, antenatal care during last pregnancy, and mother's tetanus status during last pregnancy.

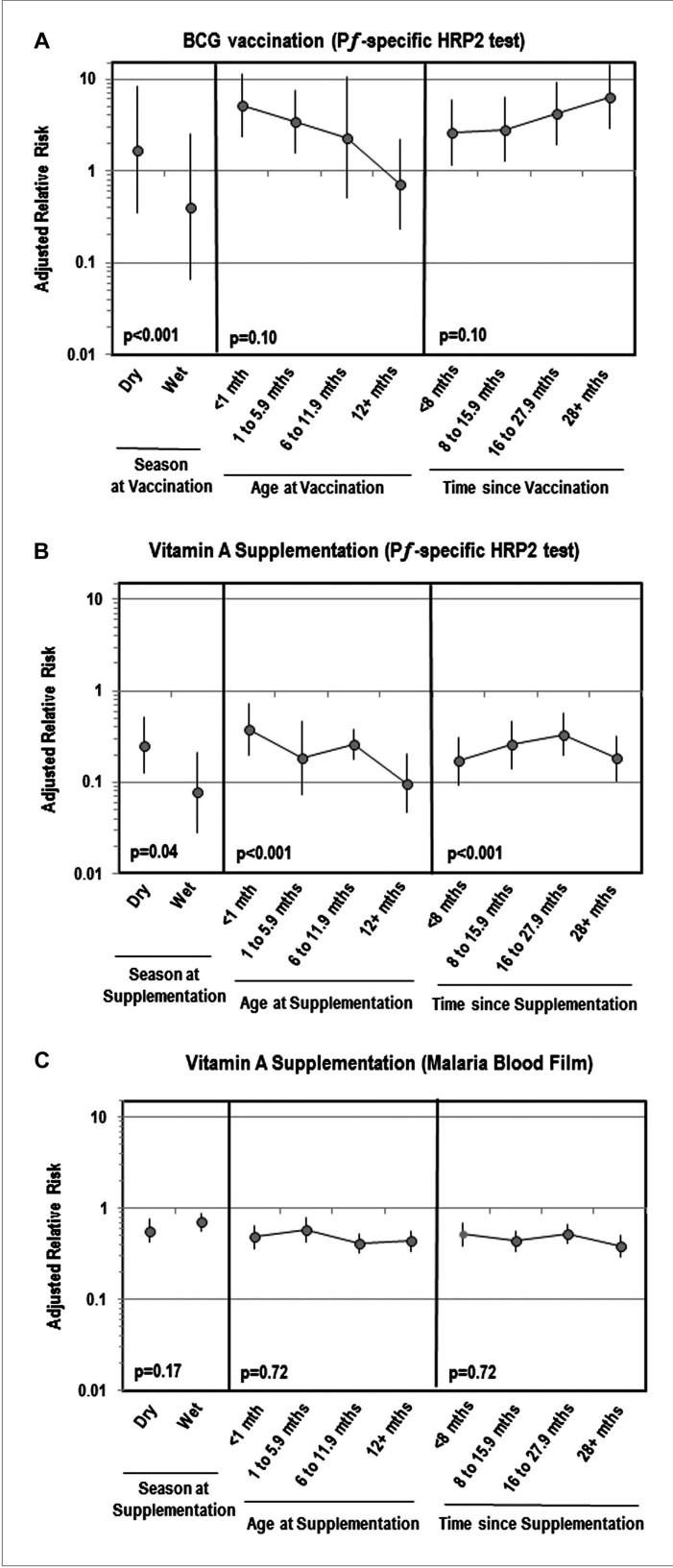

**Figure 2**. Adjusted relative risk of malaria infection according to different features of vitamin A supplementation and BCG vaccination (*Liu et al., 2012*; *World Health Organization, 2012*). (**A**) Adjusted for the following factors: age, gender, wealth index score, mother's highest level of education, malaria treatment during previous week,

*Figure 2. Continued on next page*

*Figure 2. Continued*
ownership of bed net, proportion of household members under 5 years using bed net during previous night, indoor household insecticide spraying, mother's access to antenatal care during last pregnancy, mother's knowledge regarding vertical HIV transmission, malaria transmission season, and type of community setting (urban vs rural). (**B**) Covariates 'Age at vaccination' and 'Time since vaccination' treated as continuous terms when testing for effect modification in the model. (**C**) Seasonality only available for children vaccinated in 2010 or 2011 calendar year.

association between vitamin A and *Plasmodium* parasitemia was not impacted by season, age or time since supplementation. However, for antigenemia, vitamin A was moderately more protective in older children ($p_{trend}<0.001$), the more time passed since supplementation ($p_{trend}<0.001$) or if supplementation was given during the wet season ($p_{trend}=0.04$). ***Table 3*** reveals that associations between vitamin A and parasitemia were stronger among children 36–59 months of age or if their mother took an antimalarial during pregnancy. The relation with *Pf*HRP-2 antigenemia was stronger if children had a negative malaria blood film, were 36–59 months of age, primigravidae, or had not taken an antimalarial recently. Vitamin A was more protective against *Pf*HRP-2 among children with low birth weight but was less protective against parasitemia. The association between vitamin A and *Pf*HRP-2 was stronger among children tested during the dry season or who lived in communities with a longer rainy season, and in communities with a lower annual mean for enhanced vegetation index (EVI). Associations between vitamin A and parasitemia were stronger among communities with a higher mean predicted sickle cell hemoglobin (HbS) allele frequency or median predicted glucose-6-phosphate dehydrogenase deficiency (G6PDd) allele frequency. We had insufficient data to identify cross-interactions between vaccines and vitamin A using an adjusted model, but in unadjusted models we found no significant interactions between vaccination and vitamin A supplementation.

## Discussion

In this large, population-based study within sub-Saharan Africa, vitamin A supplementation was associated with a negative risk of *Plasmodium* parasitemia and *Pf*HRP-2 antigenemia among children aged 6–59 months. None of the vaccines examined in our study (i.e. BCG, DTP, measles, and poliomyelitis) appeared to be linked with *Plasmodium* parasitemia in unweighted models but we did observe a threefold to fourfold elevated risk of *Pf*HRP-2 among BCG vaccinated children. Controlling for factors related to the utilization of vaccines, this association remained significant for BCG. Measles vaccination did not appear to be associated with either *Plasmodium* parasitemia or *Pf*HRP-2 antigenemia suggesting that the vaccine may not incur any long-term differences in the risk of malaria, as previously suggested in the literature.

### Vitamin A, DTP and gender

Our study extends previous randomized clinical trial evidence from Burkina Faso (***Zeba et al., 2008***), Papua New Guinea (***Shankar et al., 1999***), Tanzania (***Villamor et al., 2002***), and Uganda (***Nankabirwa et al., 2011***) showing consistent reductions in malaria-induced morbidity and mortality from vitamin A. Still, it contradicts suggestions that when combined with DTP (***Jorgensen et al., 2011***), vitamin A may increase the risk of malaria infection. Not only did we identify a protective association between vitamin A and malaria in a study population with high immunization coverage for BCG, DTP, and poliomyelitis, we also found in unadjusted models that DTP did not modify vitamin A's association with malaria. We found no indication of an elevated gender effect for either DTP or vitamin A, although the latter seemed to be more protective among children living in communities with higher allele frequencies for X-linked G6PDd. It is unclear how vitamin A could interact with G6PD to influence the risk of malaria infection; the only literature discussion regarding vitamin A and G6PD is limited to animal findings on adipose and lipogenic activities (***Higueret et al., 1989***; ***Arana et al., 2008***). However, G6PDd is independently related to risk reductions in malaria, a driving factor in the high prevalence of G6PDd across Africa (***Howes et al., 2013***). G6PDd is also expressed more often in boys than girls due to the X-linked nature of the gene (***Howes et al., 2013***). All of the epidemiological evidence to support the detrimental gender-based effects from vitamin A and DTP on child mortality comes from Guinea Bissau (***Benn et al., 2009a***, ***B2009b***;

**Table 3.** Modifiers of the association between vitamin A supplementation and malaria infection among children 6–59 months of age

| Characteristics at blood testing | Level in the model | *Plasmodium* species (parasitemia)* | | | | | *Plasmodium falciparum* (antigenemia)† | |
| | | No. of children with positive blood film (%) | | Unadjusted RR | Adjusted model[3] | | | |
| | | No vitamin A | Vitamin A | | RR (95% CI) | p Value (interaction term) | Adjusted RR‡ (95%CI) | p Value (interaction term) |
|---|---|---|---|---|---|---|---|---|
| **(a) Individual level** | | | | | | | | |
| **Children's characteristics** | | | | | | | | |
| Age at malaria screening | 6–35 Months | 484 (43) | 344 (20) | 0.33 | 0.46 (0.38–0.55) | <0.01# | 0.26 (0.20–0.34) | <0.01# |
| | 36–59 Months | 112 (54) | 94 (21) | 0.23 | 0.34 (0.25–0.47) | | 0.09 (0.05–0.14) | |
| Gender | Girl | 274 (44) | 226 (21) | 0.34 | 0.54 (0.44–0.67) | 0.29 | 0.23 (0.16–0.32) | 0.99 |
| | Boy | 322 (45) | 212 (19) | 0.29 | 0.40 (0.32–0.49) | | 0.23 (0.17–0.30) | |
| Pregnancy order of child | Primigravidae | 522 (47) | 366 (21) | 0.30 | 0.45 (0.38–0.53) | 0.27 | 0.20 (0.15–0.27) | 0.04 |
| | Multigravidae | 74 (31) | 72 (15) | 0.41 | 0.51 (0.39–0.66) | | 0.33 (0.22–0.48) | |
| Birth weight | 2500 mg or greater | 332 (42) | 204 (15) | 0.24 | 0.39 (0.32–0.48) | 0.01 | 0.27 (0.20–0.36) | 0.02 |
| | Less than 2500 mg | 264 (48) | 234 (28) | 0.43 | 0.53 (0.43–0.66) | | 0.13 (0.09–0.18) | |
| Treatment for intestinal worms during past 6 months | Not received | 554 (46) | 212 (25) | 0.39 | 0.50 (0.40–0.61) | 0.08 | 0.38 (0.28–0.52) | 0.07 |
| | Received | 37 (30) | 221 (17) | 0.46 | 0.66 (0.52–0.86) | | 0.15 (0.11–0.21) | |
| **Malaria-based interventions** | | | | | | | | |
| Malaria treatment during previous week | Not received | 550 (44) | 404 (19) | 0.31 | 0.44 (0.37–0.52) | 0.24 | 0.20 (0.16–0.27) | 0.02 |
| | Received | 46 (49) | 34 (33) | 0.51 | 0.88 (0.62–1.26) | | 1.01 (0.60–1.68) | |
| Mother took antimalarial during child's gestational period | No | 102 (28) | 226 (22) | 0.69 | 0.78 (0.59–1.03) | <0.001 | 0.38 (0.19–0.74) | 0.05 |
| | Yes | 491 (50) | 210 (19) | 0.23 | 0.36 (0.30–0.45) | | 0.20 (0.16–0.27) | |
| Family owns bed net | Does not own bed net | 141 (47) | 123 (22) | 0.32 | 0.43 (0.34–0.56) | 0.42 | 0.20 (0.15–0.26) | 0.70 |
| | Owns bed net | 455 (44) | 315 (41) | 0.31 | 0.48 (0.40–0.58) | | 0.24 (0.18–0.32) | |
| **(B) Community level (primary sampling unit)** | | | | | | | | |
| Type of setting | Rural | 518 (50) | 370 (25) | 0.35 | 0.46 (0.39–0.56) | 0.95 | 0.22 (0.17–0.30) | 0.91 |
| | Urban | 78 (26) | 68 (9.4) | 0.29 | 0.34 (0.22–0.51) | | 0.22 (0.12–0.40) | |
| **Genetic mechanisms of malaria protection** | | | | | | | | |
| Mean predicted HbS allele frequency§ | Less than 2.5% | 14 (20) | 104 (14) | 0.78 | 0.78 (0.32–1.91) | <0.01# | - | 0.26# |
| | 2.5–4.9% | 181 (41) | 269 (25) | 1.49 | 0.95 (0.64–1.42) | | 0.96 (0.24–3.91) | |
| | 5% or greater | 401 (48) | 65 (18) | 0.21 | 0.31 (0.17–0.56) | | 0.17 (0.08–0.39) | |
| Median predicted G6PDd allele frequency§ | Less than 7.5% | 336 (47) | 47 (11) | 0.04 | 5.42 (2.01–14.6) | <0.001# | 1.43 (0.28–7.21) | 0.02# |
| | 7.5–14.9% | 194 (43) | 128 (18) | 0.80 | 0.89 (0.52–1.50) | | 0.41 (0.12–1.34) | |
| | 15% or greater | 66 (40) | 263 (25) | 0.65 | 0.74 (0.46–1.19) | | - | |
| **Climate of communities surveyed** | | | | | | | | |
| Season of malaria transmission | Dry season | 198 (42) | 220 (19) | 0.32 | 0.40 (0.31–0.52) | 0.45 | 0.06 (0.03–0.13) | <0.001 |
| | Wet season | 398 (46) | 218 (22) | 0.32 | 0.53 (0.42–0.66) | | 0.39 (0.28–0.53) | |
| Length of rainy season | Less than 120 days | 168 (49) | 151 (21) | 0.27 | 0.43 (0.31–0.61) | <0.001# | 0.28 (0.15–0.50) | 0.03# |
| | 120–179 Days | 334 (56) | 105 (26) | 0.27 | 0.45 (0.33–0.62) | | 0.22 (0.15–0.31) | |
| | 180 Days or more | 94 (23) | 182 (17) | 0.70 | 0.77 (0.56–1.05) | | 0.21 (0.10–0.43) | |
| Length of time enhanced vegetation index above annual mean | Less than 120 days | 105 (42) | 220 (21) | 0.38 | 0.50 (0.37–0.69) | 0.18# | 0.37 (0.09–1.60) | 0.84# |
| | 120–179 Days | 486 (52) | 206 (27) | 0.34 | 0.52 (0.42–0.66) | | 0.33 (0.24–0.45) | |
| | 180 Days or more | 5 (3.3) | 12 (3.2) | 0.97 | 0.43 (0.12–1.62) | | 0.66 (0.16–2.85) | |

*Table 3. Continued on next page*

Table 3. Continued

| Characteristics at blood testing | Level in the model | Plasmodium species (parasitemia)* | | | | | Plasmodium falciparum (antigenemia)† | |
|---|---|---|---|---|---|---|---|---|
| | | No. of children with positive blood film (%) | | Unadjusted RR | Adjusted model[3] | | Adjusted RR‡ (95%CI) | p Value (interaction term) |
| | | No vitamin A | Vitamin A | | RR (95% CI) | p Value (interaction term) | | |
| Range of enhanced vegetation index per year | Less than 0.20 | 46 (25) | 41 (7.9) | 0.26 | 0.38 (0.22–0.63) | 0.18# | 0.24 (0.13–0.47) | 0.13# |
| | 0.20–0.29 | 247 (44) | 105 (17) | 0.25 | 0.48 (0.36–0.64) | | 0.36 (0.25–0.53) | |
| | 0.30 or greater | 303 (51) | 292 (28) | 0.38 | 0.48 (0.39–0.61) | | 0.19 (0.13–0.29) | |
| Annual mean for enhanced vegetation index | Less than 0.20 | 150 (37) | 21 (7.2) | 0.13 | 0.17 (0.11–0.28) | <0.01# | 0.16 (0.10–0.25) | 0.01# |
| | 0.20–0.29 | 309 (59) | 89 (24) | 0.22 | 0.51 (0.37–0.70) | | 0.34 (0.23–0.51) | |
| | 0.30 or greater | 137 (33) | 328 (22) | 0.56 | 0.61 (0.46–0.81) | | 0.56 (0.20–1.53) | |

HRP-2: histidine rich protein 2; RR: relative risk; CI: confidence interval; na: not applicable: ref: reference category; HbS: hemoglobin S; G6PD: glucose 6-phosphate dehydrogenase deficiency.

*Tested in four countries: Burkina Faso, Mozambique, Rwanda and Senegal.

†Tested in three countries: Burkina Faso, Rwanda and Senegal.

‡Adjusted for the following factors: age, gender, wealth index score, mother's highest level of education, malaria treatment during previous week, ownership of bed net, proportion of household members under 5 years using bed net during previous night, indoor household insecticide spraying, mother's access to antenatal care during last pregnancy, mother's knowledge regarding vertical HIV transmission, malaria transmission season, and type of community setting (urban vs rural).

§Geographical waypoints were not recorded for 40 communities. Subjects from these PSUs were excluded from analysis.

#Covariate treated as continuous term when testing for effect modification in the model.

Fisker et al., 2013). Whether relatively mild phenotypic differences between G6PDd alleles could explain such different vaccine effects is not clear.

## Placental malaria and vitamin A

Differences in the impact of vitamin A on malaria infection by birth weight, and having a mother who used antimalarials during the child's gestational period suggests exposure to placental malaria may play a critical role in vitamin A's protective effect mechanisms later on in childhood. Published evidence regarding placental malaria, antenatal vitamin supplementation, and infant health are limited. However, Cox et al. (2005) have identified a lower level of antibodies involved in placental parasite sequestration and risk reductions in active placental malaria infection at delivery in mothers who received vitamin A supplements during pregnancy. Additional research may be warranted on the impact of vitamin A supplementation in young children with known prenatal exposure to malaria.

## BCG and soil-transmitted helminths

Results from a previous clinical trial in Guinea Bissau provides no evidence of an association between BCG re-vaccination and malaria parasitemia (Rodrigues et al., 2007). Although the study had a number of design limitations (i.e. did not examine the magnitude of the re-vaccination boosting effect on prior immunity from BCG, was statistically underpowered (potentially causing type I error), and based on passive case detection of malaria at health centers/outpatient clinics), it does suggest that our inconsistent findings between Plasmodium parasitemia and PfHRP-2 antigenemia may be due to another factor. In 2012, the tropical diseases research (TDR) programme reported false positivity caused by schistosomiasis and Chagas in one lot of Paracheck Pf rapid diagnostics (World Health Organization on behalf of the Special Programme for Research and Training in Tropical Diseases, 2011). Infection with Trypanosoma cruzi, schistosomiasis, and other soil-transmitted helminths is inversely associated with anti-mycobacterial responses due to Th1/Th2 polarization (Malhotra et al., 1999; Elias et al., 2005; Brown et al., 2006; Dauby et al., 2009), and albendazole increases BCG immune reactivity among children infected with schistosomiasis (Dauby et al., 2009). Although Chagas is not endemic to the countries we studied (Morel and Lazdins, 2003), there is growing evidence that

schistosomiasis may be an overlooked, endemic disease among infants and young children in sub-Saharan Africa (*Russell Stothard et al., 2013*).

## Strengths, limitations and additional research direction

Our study represents an active population-based surveillance of malaria infection within four countries. The breadth and size of the study population enabled us to evaluate the association between malaria and childhood vaccines/vitamin A despite high vaccination coverage rates within most of the communities surveyed (*Desai et al., 2005*). Although our models adjusted for factors that could explain study population differences between blood film and antigenemia models, to address this issue further we ran a sensitivity analysis using blood film data including/excluding Mozambique. This analysis showed that when Mozambique data were excluded from blood film analysis, the magnitude of protective vitamin A effects in both blood film and antigenemia models were nearly identical. Blood films and antigenemia readings are used to identify two different outcomes (i.e. blood films test for *Plasmodium* spp. while antigenemia test for one particular type of *Plasmodium* [*P. falciparum*]). However, due to geography, blood films were likely to be detecting *P. falciparum* due to geography (*Gething et al., 2011, 2012*). This would explain the similar results identified between blood film and antigenemia results when excluding Mozambique data from our analysis. We have chosen to present data in this paper that includes Mozambique because these results provide more conservative estimates of vitamin A effects when using blood films. A number of study limitations should also be considered when reviewing our results.

First, while we used propensity scores to address differences in patient characteristics associated with vaccine uptake, the possibility of residual confounding due to unmeasured factors remains. Moreover, because of the observational nature of our study design, high-quality randomized clinical trials are needed to confirm the efficacy of vitamin A in preventing malaria, particularly among children exposed to placental malaria. This includes further clarification regarding optimal dosages and the time sequence in which vitamin A is administered in relation to other vaccines, as well as more accurate measures with regard to HbS and G6PDd allele frequency. In particular, our study only considered information regarding the last recorded dose of vitamin A supplement rather than a child's history of supplementation. Given that vitamin A is fat soluble and can be stored in the liver for long periods of time (*Blomhoff, 1994*; *Tanumihardjo, 2011*), additional research is needed to examine whether regular supplementation has an impact on increased protection.

Second, although the EVI dataset is advantageous for this research due to its spatial and temporal resolutions, several caveats are associated with EVI that affect its utility as a proxy for seasonal rainfall. Foremost among these concerns is the time lag between rainfall and vegetation response, as dormant vegetation does not immediately become green following the return of seasonal rains (*Nicholson et al., 1990*; *Davenport and Nicholson, 1993*). This issue will create a temporal offset between EVI and rainfall that may complicate the interpretation of the results of this research. However, this concern will be partially mitigated by a similarly lagged response to rainfall in populations of vector species responsible for spreading *Plasmodium* (*Mbogo et al., 2003*; *Galardo et al., 2009*). Another concern when using EVI as a proxy for rainfall relates to non-uniform responses of vegetation to rainfall that reflect long-term moisture conditions (via land cover) rather than seasonal oscillations (*Townshend and Justice, 1986*). This issue is most apparent when comparing dry season EVI values in grasslands to those in forests, as forests tend to retain more green vegetation (and thus have higher EVI values) even when dry due to deeper root systems, etc. A second aspect of EVI linked to land cover differences is the speed at which vegetation responds to rainfall, as plants in drier ecosystems have evolved to respond more quickly to infrequent rain events (*Ogle and Reynolds, 2004*). For this research, differing responses to land cover are accounted for by deriving unique EVI curves for each survey cluster location, thus creating relative metrics based on only localized (i.e. per-cell) EVI values rather than regional summaries that contain responses of multiple land cover types.

Finally, data on children's HIV status were not available. Although the possibility of mother-to-child HIV transmission was low in our study (less than 1% of mothers were HIV positive), HIV is an established risk factor for malaria infection (*Abu-Raddad et al., 2006*). To address this issue, we adjusted a subgroup of models for maternal HIV status. While the protective association between vitamin A and malaria remained, the low prevalence of maternal HIV in our study population (and hence low statistical power) makes it difficult to rule out potential vitamin A–HIV interactions. Further research is needed to evaluate potential interactions between vitamin A, HIV and malaria infection.

## Materials and methods

### Study design and data collection

Data were extracted from Macro International Demographic and Health Surveys (DHS) (*ICF International, 2013*), a database of nationally representative household surveys conducted in low and middle-income countries. We focused on standard DHS surveys carried out since January 2010, which tested children for malaria using blood films (gold standard malaria test) and documented children's use of standard vaccines and vitamin A supplements. Malaria indicator surveys were excluded from our study due to the absence of data on childhood immunization. From 20 country surveys with accessible data (last checked 15 January 2014), we identified four surveys that fitted these criteria (i.e. Burkina Faso, Mozambique, Rwanda, and Senegal). All surveys were conducted between May 2010 and November 2011, and used a multistage sampling process, by which study participants were identified from randomly selected households within a primary sampling unit (PSU) (e.g. census enumeration tract). Trained interviewers used standardized questionnaires to collect information from participants during home interviews on a range of issues related to population, health, and nutrition. Geographical waypoints (World Geodetic System 84 datum, latitude, and longitude) were also recorded at the center of each PSU. Eligible subjects for our analysis included children 6–59 months of age who, during survey interviews, provided blood samples for malaria screening and had their history of vaccination and vitamin A supplementation documented based on health records. Each DHS had comprehensive ethical approval and written informed consent was collected for each survey participant (see below). Additional data were extracted from external sources on malaria seasonality and the population prevalence of malaria protective genes using the geographical coordinates of each PSU (n=2604 community waypoints).

### Data on climate conditions for malaria

To estimate seasonality, an EVI dataset was used as it relates to seasonal precipitation, albeit lagged in time, and the emergence of vector mosquito species. This dataset was created from moderate resolution imaging spectroradiometer (MODIS) bidirectional reflectance distribution function (BRDF)-corrected composites (MCD43B4) (*Schaaf et al., 2002*), which have a 1 km spatial resolution and a 16-day temporal resolution. By utilizing MODIS data from sensors located on both the Aqua and Terra satellite platforms, the temporal resolution of the final EVI dataset was increased to 8 days, or 46 grids (i.e. rasters or images) per year. The MODIS BRDF data were acquired in tiles from the NASA Reverb site (http://reverb.echo.nasa.gov/), 42 of which were required to create each 8-day mosaic of Africa for 2000 through 2012. EVI was then calculated for each mosaic using the equation defined by *Huete et al. (1999)*. Due to cloud cover, which was particularly problematic in forested regions of equatorial west Africa, a cloud-filling algorithm was applied to each EVI grid to create a spatially complete dataset (*Weiss et al., 2014*). From the gap-filled EVI dataset, values were extracted in R 3.0.0 (R Foundation, Vienna, Austria) for each PSU waypoint for all dates from 2010 and 2011 to create a longitudinal EVI profile for each point. From each profile, we calculated the following indices for both years of the study: i. mean EVI; ii. start of the rainy season corresponding to an EVI above the annual mean; iii. end of the rainy season, corresponding to an EVI below the annual mean occurring after ii; iv. number of days between ii and iii; v. number of days with an EVI above the annual mean; vi. date of the minimum EVI; and vii. date of the maximum EVI.

### Data on genetic mechanisms of malaria protection

Predicted allele frequencies for HbS (*Piel et al., 2013*) and G6PDd (*Howes et al., 2012*) were extracted for each PSU waypoint from continuous global predicted surface maps of the worldwide distribution of these genes using ArcGIS 10.1 (Esri, Redlands, CA). These surface maps were generated as part of the Malaria Atlas Project (MAP) (*Malaria Atlas Project, 2013*), using Bayesian geostatistical models of prevalence data collected from a systematic review of community-based surveys (*Gething et al., 2011*). The survey database consisted of both published and unpublished literature (*Moyes et al., 2013*).

### Identification of vaccination and vitamin A supplementation history

Information regarding children's vaccination and vitamin A supplementation history was ascertained from health card data recorded during the survey interview. This included a child's date of vaccination using BCG, DTP, measles and polio as well as the number of vaccine doses received (for DTP and polio only). We used these data then to calculate age in months at the moment of vaccination, and time in months since vaccination. Children whose mother/guardian responded in the affirmative to vaccination

(e.g. child vaccinated during campaign) but did not provide health cards to document this information were excluded from the analysis. For vitamin A supplementations, data were recorded regarding the date of last dose received and use of the vitamin supplement during the 6 months prior to the survey interview.

### Determination of malaria status

Primary study outcomes included the microscopic detection of *Plasmodium* spp. to assess parasitemia and the presence of *Pf* HRP-2 to assess antigenemia. The microscopic presence of *Plasmodium* spp. was determined by reading Giemsa-stained thick blood films. Species of *Plasmodium* spp. were not recorded. Capillary blood was collected during household interviews and tested onsite for malaria antigenemia. *Pf*-specific HRP-2 antigens were detected using the rapid diagnostic Paracheck *Pf* (Orchid Biomedical Services, Goa, India) for Burkina Faso and Senegal surveys and first response malaria antigen HRP-2 (Premier Medical Corporation, Daman, India) for the Rwanda survey. Thick blood smears were prepared by trained technicians for all children, and sent for gold standard testing of malaria to the Centre National de Recherche et de Formation sur le Paludisme (Burkina Faso), Centro de Investigações em Saúde de Manhiça (Mozambique), TRAC/Plus Malaria Unit in Kigali (Rwanda) and Department of Parasitology at the Université Cheikh Anta Diop (Senegal). Internal quality control was conducted using standard laboratory protocol and procedures, including having slides randomly read by different laboratory technicians or the chief technician/supervisor for internal quality control measures. External quality control (Mozambique only) included selecting samples by a computerized system developed by ICF Macro and sending them for external quality control testing at the Muraz Center (Bobo-Dioulasso).

### Determination of current HIV status among mothers

Capillary blood was collected during household interviews from all women aged 15–29 years. Samples were collected on filter paper, dried for 24 hr and then sent for testing to the Centre Regional de Transfusion Sanguine de Ouagadougou (Burkina Faso), National Reference Laboratory (Rwanda) and National Reference Laboratory of Bacteriology and Virology at A Le Dantec Hospital (Senegal). Viral antibodies were detected using the enzyme-linked immunosorbent assay (ELISA)-based assay VironostikaÒ HIV Uni-Form II plus O (Biomériux, Marcy l'Etoile, France). Positive samples and a portion of negative samples were tested using the ELISA assay EnzygnostÒ Anti-HIV ½ plus (Siemens AG, Erlangen, Germany). Discordant samples were further tested using the Pepti Lav assay (Bio-Rad Laboratories, Hercules, CA) or InnoLia (Burkina Faso and Rwanda). Internal quality control measures (Burkina Faso, Rwanda and Senegal) included using control wells (positive and negative) provided with manufacturer's screening kit on each test plate. All positive samples and 10% of randomly selected negative samples were also tested using more than one assay, with discordant samples further tested using a third assay. External quality control measures (Burkina Faso only) consisted of sending all positive samples and 80 randomly selected negative samples to the Center Muraz (Bobo-Dioulasso) for additional HIV testing.

### Statistical analysis

Multivariable logistic regression models were used to estimate associations between malaria and childhood vaccination/supplementation. The impact of age at vaccination, time since vaccination, and malaria season on the date of vaccination were also examined. To avoid statistical inference errors in our results due to clustering and stratification from the sampling process, we defined country and PSU as strata, and households as clusters using logistic regression models that accounted for the complex survey data structure. To test for linear trends, covariates were included as continuous terms in the regression model. We adjusted models for continuous variables of age and wealth index score, and categorical variables of gender (boy vs girl), malaria treatment during previous week (yes vs no), ownership of bed net (yes vs no), proportion of household members under 5 years old using bed net during previous night (none, all, some, or no bed net), indoor household insecticide spraying (yes vs no), malaria transmission season (dry vs wet), community setting (urban vs rural), mother's highest educational level (none, incomplete primary, complete primary, incomplete secondary, complete secondary, post-secondary), and access to antenatal care during last pregnancy (yes vs no). Complete subject analysis was used. *Supplementary file 1* presents children's characteristics for missing (excluded) and non-missing (included) groups.

Effect modifiers for each vaccine/vitamin model were identified by individually testing cross-product terms between type of immunization/supplement received and individual-based characteristics including age, gender, pregnancy order of child, birth weight, receipt of antimalarial during previous week, receipt of intestinal worm treatment during previous 6 months, if mother took antimalarial during child's gestational period, family owns bed net, or type of community setting; climate-related factors including malaria transmission season at time of immunization, length of rainy season, length of time that EVI was above annual mean, range of EVI per year, or annual mean for EVI; and malaria genetic factors including the predicted allele frequency in a community for HbS (mean estimate) and G6PDd (median estimate). To account for uncertainty in spatial prediction values for HbS and G6PDd, we conducted a sensitivity analysis by excluding from models any prediction value with degrees of uncertainty (i.e. interquartile range) greater than 20%.

## Avoidance of selection bias

Due to the observational nature of the data (not randomized), we controlled for differences in patient characteristics associated with vaccine or vitamin use, by inverse probability weighting of models using the propensity score for receiving a vaccine or vitamin supplement. Predictors of vaccination/supplementation included in propensity score models were: age, gender, low birth weight, presence of radio/television in home, urban versus rural setting, breastfeeding status, wealth index score, as well as mother's age, highest education level, antenatal care during last pregnancy, and tetanus status during last pregnancy. Standardized bias measures were used to assess numerical balance in propensity score covariate distributions, before and after IPW weighting (*Supplementary file 2*). Factors with standardized bias values ≤0.25 were considered balanced.

All statistical analyses were completed using SAS v.9.3 (SAS Institute, Cary, NC) and R v.3.0.0 (R Foundation, Vienna, Austria).

## Ethical considerations

Prior to enrolling in the survey, the parent/adult responsible for the child had to provide written informed consent to participate, along with additional informed consent for blood sample collection and testing. If a child tested positive during the survey using malaria RDT, their parent/guardian was informed of results and the child was immediately given treatment according to the current treatment guidelines. HIV tests for adults were anonymous, with survey data only being linked to test results after respondent identifiers were deleted from the database. Although HIV test results were not available to study participants, patients were offered cards enabling them to obtain free HIV testing and counselling at Voluntary Testing Centers. The Institutional Review Board of ICF Macro reviewed and approved the MEASURE Demographic and Health Surveys Project Phase III, in compliance with United States (U.S.) Department of Health and Human Services (DHHS) regulation 45 Code of Federal Regulations (CFR) 46 for 'Protection of Human Subjects' research. Protocols for blood specimen collection, HIV and malaria testing were also approved by ICF Macro International Institutional Review Board (IRB) [all countries] in compliance with U.S. DHHS regulation 45 CFR 46 for 'Protection of Human Subjects' research, National Ethics Committees [Burkina Faso, Rwanda and Senegal] and U.S. Centers for Disease Control [Rwanda only]. For Senegal, supervisory visits were also organized by the National Ethics Committee to ensure field compliance with ethical regulations. Geographical waypoints were randomly displaced to ensure confidentiality [Urban PSU: positional error of 0–2 km; and, Rural PSU: positional error of 0–5 km (99% of clusters), or 0–10 km error (1% of clusters)]. DHS provided anonymized data to study authors. As a secondary analysis of publically available de-identified data, the study was determined exempt from ethics review by Johns Hopkins Bloomberg School of Public Health IRB Office according to U.S. DHHS regulation 45 CFR 46.102 for non-human subject research.

## Acknowledgements

The authors wish to thank Dr David Sullivan for helpful conversations regarding malaria diagnostics, Elizabeth Alden for comments on an earlier draft, Bridgette Wellington, Lia Florey and Dean Garrett for technical information related to DHS surveys, Josh Colston for organizing data access, and Rachel Upton for administrative assistance. MGHD is funded by a Post-doctoral Training Award from the Fonds de la Recherche Québec en Santé (#24615). DJW receives support from the Bill and Melinda Gates Foundation (#OPP1068048). SIH is funded by a Wellcome Trust Senior Research Fellowship (#095066), which also

supports REH. SIH also receives support from the Bill and Melinda Gates Foundation (#OPP1106023). This work was presented in part at the 62nd Annual Meeting of the American Society of Tropical Medicine and Hygiene, Washington, DC, November 13–17, 2013 (Abstract #LB-2124).

## Additional information

### Competing interests

SIH: Reviewing editor, *eLife*. The other authors declare that no competing interests exist.

### Funding

| Funder | Grant reference number | Author |
| --- | --- | --- |
| Bill and Melinda Gates Foundation | OPP1068048 | Daniel J Weiss |
| Fonds de Recherche du Québec—Santé | Post-doctoral Training Award, 24615 | Maria-Graciela Hollm-Delgado |
| Wellcome Trust | Senior Research Fellowship, 095066 | Simon I Hay |
| Bill and Melinda Gates Foundation | OPP1106023 | Simon I Hay |

The funders had no role in study design, data collection and interpretation, or the decision to submit the work for publication.

### Author contributions

MGHD, FBP, DJW, REH, Conception and design, Acquisition of data, Analysis and interpretation of data, Drafting or revising the article; EAS, REB, Conception and design, Analysis and interpretation of data, Drafting or revising the article; SIH, Conception and design, Acquisition of data, Drafting or revising the article

### Author ORCIDs

Simon I Hay, http://orcid.org/0000-0002-0611-7272

### Ethics

Human subjects: The Institutional Review Board of ICF Macro reviewed and approved the MEASURE Demographic and Health Surveys Project Phase III, in compliance with United States (U.S.) Department of Health and Human Services (DHHS) regulation 45 Code of Federal Regulations (CFR) 46 for 'Protection of Human Subjects' research. Protocols for blood specimen collection, HIV and malaria testing were also approved by ICF Macro International Institutional Review Board (IRB) [all countries] in compliance with U.S. DHHS regulation 45 CFR 46 for 'Protection of Human Subjects' research, the National Ethics Committees [Burkina Faso, Rwanda and Senegal] and U.S. Centers for Disease Control [Rwanda only]. DHS provided anonymized data to study authors. As a secondary analysis of publically available de-identified data, the study was determined exempt from ethics review by Johns Hopkins Bloomberg School of Public Health IRB Office according to U.S. DHHS regulation 45 CFR 46.102 for non-human subject research.

## Additional files

### Supplementary files

• Supplementary file 1. Comparison of subject groups with complete and missing information on confounders.

• Supplementary file 2. Comparison of standardized bias for factors associated with vaccine/vitamin A supplement uptake, before and after inverse probability weighting (IPW).

• Supplementary file 3. Overview of immunization schedules for each country.

## Major dataset

The following previously published dataset was used:

| Author(s) | Year | Dataset title | Dataset ID and/or URL | Database, license, and accessibility information |
|---|---|---|---|---|
| ICF International | 2013 | Demographic and Health Surveys (various) | http://www.dhsprogram.com/ | Publicly available. |
| Malaria Atlas Project (MAP) | 2013 | Malaria Atlas Project (MAP) | http://www.map.ox.ac.uk/ | Publicly available. |

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
