## [Decision Letter]

Thank you for sending your work entitled “Vitamin A supplements, routine
immunization, and the subsequent risk of malaria among children 6 to 59 months of
age” for consideration at *eLife*. Your article has been evaluated
by Prabhat Jha (Senior editor), a Reviewing editor, and 4 reviewers.

The following individuals responsible for the peer review of your submission have agreed
to reveal their identity: Stephen Tollman (Reviewing editor) and John Pettifor (peer
reviewer). Please note that a further three reviewers conducted detailed
assessments.

The Reviewing editor and the reviewers discussed their comments before we reached this
decision, and the Reviewing editor has assembled the following comments to help you
prepare a revised submission.

The study aimed to assess the effects of both vitamin A supplementation and immunization
on malaria parasitemia and antigenemia in sub–Saharan children between the ages
of 6 and 59 months, in order to verify the findings in experimental animals in which
vitamin A supplementation and immunizations have been shown to confer some protection
against malaria infection. The authors conclude that BCG vaccination did not influence
malaria parasitemia but did increase PƒHRP2 antigenemia, while vitamin A
supplementation reduced the prevalence of both parasitemia and antigenemia. The BCG
effect on antigenemia was thought to be due to an increase in false positivity due to
Chagas disease or *Schistosomiasis*. Due to the low malaria burden in two
of the four countries, the study is driven by data from Burkina Faso and Mozambique. If
confirmed, the findings are of importance to the many children living in high malaria
infection zones—with potential policy impact.

Substantive comments are detailed below. However, the major concern of all reviewers
relates to limitations of the DHS dataset, ensuring that comparisons and analyses of the
observational work are valid and adequately explained/justified, gaining a better
understanding of country–level effects, and concerns about ethical practice in
global health survey work.

Major/substantive comments:

1) The four populations studied differ in malaria burden, climate and vitamin A use. It
is unclear whether some of the differences between the blood film and RDT assessments
are due to the omission of Mozambique (one of two countries with relatively high
parasite prevalence) from the latter. Further, amidst the range of
correcting/confounding variables it is not clear how 'country' as a
geopolitical unit was considered.

2) Because the populations are not the same for blood film and antigenaemia readings, it
is hard to interpret the findings from the two analyses (e.g. for vitamin A, where
age–and time–dependent trends with antigen are presented but are difficult
to explain biologically, unless they reflect host and environmental/social differences
between Burkina Faso, which contributes the overwhelming number of malaria antigen
cases, and Mozambique). This makes it difficult to single out the take–home
message; or have confidence that the antigen findings are generalisable.

3) Clearly there is value in capitalising on the cross–country comparisons
possible from DHS data. But the Methods section could be much clearer on whether
children vaccinated were compared with those unvaccinated (and if not, how the groups
assessed for protection were selected). Since this is observational rather than trials
data, adequate explanation of this design/analysis issue is essential. (High vaccination
coverage overall raises the question whether comparing vaccinated against unvaccinated
children is valid given possible selection biases.)

4) Quality of National Data: The authors are silent on country–level quality
control activities for malaria diagnosis, especially on the expertise and methods used
by those responsible for blood smear readings.

5) Ethical approval: The authors state that “the study was determined exempt from
ethics review by Johns Hopkins Bloomberg School of Public Health”. It is not
appropriate to have such approval from Johns Hopkins only; an understanding of the
ethical status of the work in countries where the data were collected is needed.
Furthermore, it would be helpful to know how children who tested malaria positive were
managed.

6) It is mentioned that 20,948 children presented health cards during the survey
interviews. It is stated that 12,028 were tested using thick blood films and 8702 using
RDTs, but were the tests on each child limited to either thick film examination or RDT
or both? If only one, how was concordance determined; and if both, what was the overlap
(and please explain how children were accounted for as either thick film (12,028) or RDT
(8702) to sum to 20,948)?

Note: numbers in the text and Tables 1 and 2 are not all in agreement. Since microscopy is the stated 'gold
standard' it might be clearer if the analysis in Table 1 included only those children with microscopy results.

7) Please provide information on selection of confounders. And consider:

a) HIV status and vertical HIV exposure, as these may influence a child's immune
response to both vaccination and vitamin A supplementation;

b) Number of doses of each of the immunizations or vitamin A supplements received (it is
likely that older children had more immunizations and A supplements than younger). Were
immunization schedules and supplementation programmes similar in the four countries?

8) Third paragraph in the Results section: the authors indicate that vitamin A was more
protective the greater the time from the last dose. The last panel of Figure 2 does not seem to show this, nor does the
statement tally with the first paragraph of the Discussion.

9) Fourth paragraph in the Discussion section: the authors suggest that the increase in
antigenemia associated with BCG vaccination might be due to false positivity from other
parasitic infections; also that false–positivity has been reported in Paracheck
assays. However, given low numbers and wide CIs, a type I error could be a possibility.
Also, since different assays were used, can the authors advise whether the increase in
antigenemia was similar in the subsamples when using either one of the assays?

Please note that Chagas disease is not found in these countries. As most children tested
were very young, they are unlikely to have significant exposure and antibodies to
*Schistosomiasis*.

---

## [Author Response]

*1) The four populations studied differ in malaria burden, climate and vitamin A
use. It is unclear whether some of the differences between the blood film and RDT
assessments are due to the omission of Mozambique (one of two countries with
relatively high parasite prevalence) from the latter. Further, amidst the range of
correcting/confounding variables it is not clear how 'country' as a
geopolitical unit was considered*.

We thank the reviewers for their insightful comment and recognize their concerns. To
address the issue of study population differences, we ran a sensitivity analysis using
blood film data including and excluding Mozambique from our analysis. This analysis
(Author response table 1) showed that when excluding Mozambique data, vitamin A not only
remained protective but the magnitude of its protective effect was amplified. In other
words, including data from Mozambique in the blood film model, provided a more
conservative estimate of vitamin A effects. We have included the more conservative
results in our paper. Unfortunately, RDT testing was not conducted among Mozambique
subjects making it impossible to conduct similar sensitivity analysis for RDT
testing.

Our models did adjust for factors that could explain differences between the four
populations (including vitamin A use and climate). This is indicated in the seventh
paragraph of the Materials and methods section of the manuscript. Furthermore, to avoid
statistical inference errors in our results due to clustering and stratification from
the sampling process, we used logistic regression models that accounted for complex
survey data structure and defined country and primary sampling unit (∼district)
as strata, and household as cluster. We recognize that this was not fully clear in our
original paper and have now incorporated this information in the seventh paragraph of
the Materials and methods section of the manuscript.

Author response table 1. Adjusted PS-IPW relative risk of malaria, by group of children
analyzedType of Vaccine*Adjusted RR (95%CI)**Plasmodium spp.* Parasitemia (Microscopic
Examination of Blood Film)*Plasmodium falciparum* Antigenemia (Rapid
Detection of HRP2) (i.e., excl. Mozambique)All Children (i.e., incl. Mozambique)Children from countries using both film & RDT (i.e., excl.
Mozambique)*BCG*1.24 (0.76-2.05)2.71 (1.36-5.39)3.52 (1.66-7.48)*DTP*0.06 (0.01-0.47)0.38 (0.01-10.1)0.06 (0.01-0.38)*Measles*1.01 (0.20-5.19)1.00 (0.18-5.45)0.68 (0.15-3.12)*Polio*0.74 (0.27-2.01)0.60 (0.22-1.62)0.93 (0.37-2.35)*Vitamin A***0.43 (0.36-0.52)****0.24 (0.18-0.34)****0.22 (0.16-0.29)***Countries included in analysis**Burkina Faso, Rwanda, Senegal & Mozambique**Burkina Faso, Rwanda, Senegal*

Abbreviations: PS IPW, Propensity-score inverse probability weighting

*2) Because the populations are not the same for blood film and antigenaemia
readings, it is hard to interpret the findings from the two analyses (e.g. for
vitamin A, where age–and time–dependent trends with antigen are
presented but are difficult to explain biologically, unless they reflect host and
environmental/social differences between Burkina Faso, which contributes the
overwhelming number of malaria antigen cases, and Mozambique). This makes it
difficult to single out the take*–*home message; or have
confidence that the antigen findings are generalisable*.

We apologize for the confusion. As previously mentioned, our models was adjusted for
factors that could explain population differences. More importantly, our analysis in
Author response table 1 shows that when Mozambique data was excluded from the blood film
analysis (i.e., populations were identical for blood film and antigenaemia results), the
magnitude of protective vitamin A effects increased causing results in both the blood
film and antigenemia models to be nearly identical.

Technically speaking, blood films and antigenaemia readings are used to identify two
different outcomes. Blood films test for *Plasmodium* spp., while
antigenemia is testing for one particular type of *Plasmodium
(P.falciparum).* We recognize the confusion caused by using these two terms,
and have tried to make the distinction between these tests clearer by re-labeling tables
and figure using taxonomic terms for malaria, restructuring Tables 2 and 3, as well as referring to type of
*malaria sp*. detection throughout the manuscript. However, it should
be noted that due to geography, blood films were likely detecting
*P.falciparum*. This would explain the similar results identified
between blood film and antigenemia results when excluding Mozambique data from our
analysis.

That said, we have chosen to include Mozambique data for blood films in our manuscript
since these results provide more conservative estimates of Vitamin A effects. All of
this discussion is now included in the fifth paragraph of the Discussion section in the
manuscript.

*3) Clearly there is value in capitalising on the
cross*–*country comparisons possible from DHS data. But the
Methods section could be much clearer on whether children vaccinated were compared
with those unvaccinated (and if not, how the groups assessed for protection were
selected). Since this is observational rather than trials data, adequate explanation
of this design/analysis issue is essential. (High vaccination coverage overall raises
the question whether comparing vaccinated against unvaccinated children is valid
given possible selection biases*.*)*

We fully agree with the reviewers regarding the value of cross-country comparisons. In
fact, during the initial stages of our analysis we had hoped to present
country-stratified models, but were unable to do so due to issues with model
convergence. A similar problem occurred when including country as a covariate in
‘overall’ models investigating effect modification. As for the
observational nature of the study, we recognized early on that vaccination uptake in
itself might introduce a selection bias. To account for this, we used inverse
probability weighting by propensity scores. The propensity score was based on known
predictors for use of vaccination. We recognize that this may not have been fully clear
in our manuscript, and have now included this information under the header
“Avoidance of Selection Bias” in the Materials and methods section of the
manuscript. Please note that the unweighted models (which did not account for potential
vaccination selection bias) provided more conservative estimates, and for this reason
are included in the paper.

*4) Quality of National Data: The authors are silent on
country*–*level quality control activities for malaria
diagnosis, especially on the expertise and methods used by those responsible for
blood smear readings*.

We are grateful for the reviewers’ comment. Information on both internal and
external quality control measures for malaria testing is now included in the fifth
paragraph of the Materials and methods section. We have also included quality control
measures used for HIV testing in the same section.

*5) Ethical approval: The authors state that “the study was determined
exempt from ethics review by Johns Hopkins Bloomberg School of Public Health”.
It is not appropriate to have such approval from Johns Hopkins only; an understanding
of the ethical status of the work in countries where the data were collected is
needed. Furthermore, it would be helpful to know how children who tested malaria
positive were managed*.

A specific section entitled ‘Ethical Considerations’ has now been added to
the paper, describing detailed information on ethical procedures related to the study.
Please note that, aside from ethics review by JHSPH IRB, additional review was conducted
during the original DHS survey by the ICF Macro International IRB (all four countries),
the National Ethics Committees (three countries), and by the U.S. Centers for Disease
Control (one country). Also, if a child tested positive during the survey using malaria
RDT, his/her parent/guardian was informed of the results and the child was immediately
given treatment according to the current treatment guidelines. This information is now
included in the “Ethical Considerations” paragraph in the manuscript.

*6) It is mentioned that 20,948 children presented health cards during the survey
interviews. It is stated that 12,028 were tested using thick blood films and 8702
using RDTs, but were the tests on each child limited to either thick film examination
or RDT or both? If only one, how was concordance determined; and if both, what was
the overlap (and please explain how children were accounted for as either thick film
(*[54]*) or RDT (8702) to sum to 20,948)?*

*Note: numbers in the text and*
Tables 1 and 2
*are not all in agreement. Since microscopy is the stated 'gold
standard' it might be clearer if the analysis in*
Table 1
*included only those children with microscopy results*.

We apologize for the confusion. From 20,948 children who presented health cards during
survey interviews, 18,413 were eligible for blood screening from which 12,058 provided
capillary blood for malaria testing. Figure 3 shows that 99.8% of children were tested with blood film, and more
than 70% were tested using both blood film and RDT. Less than 0.5% were tested using
only RDT. The 12,028 tested using thick films and 8702 tested using RDT’s come
from the 12,058 subjects providing blood, and not the 20,948 children who presented
health cards. We recognize that the flow chart (Figure 1 in our original paper) may not have been clear on this point, and have now
included a statement describing the breakdown of diagnostic tests used, in the Results
section of the manuscript. Also, as requested by reviewers, Table 1 now only includes children with blood film microscopy
results.Author response image 1.Proportion of children tested according to type of diagnostic used.

*7) Please provide information on selection of confounders. And consider: a) HIV
status and vertical HIV exposure, as these may influence a child's immune
response to both vaccination and vitamin A supplementation*;

We appreciate the potential impact of HIV on associations between vitamin A and malaria.
Unfortunately, data on children’s HIV status was not available. However, 2540
mothers of children participants in Burkina Faso, Rwanda and Senegal were tested by the
DHS for HIV status. Using this data, we found less than 1% of mothers tested positive
for HIV, suggesting a low potential for vertical transmission in our study population.
This information is included in the first paragraph of the Results section. Adjusting
for maternal HIV status in models using subjects whose mothers were tested, we found
that the association between vitamin A and malaria remained protective (Author response
table 2). Still, the low prevalence of maternal HIV in our study population (and hence
low statistical power) makes it difficult for us to rule out potential vitamin
A*HIV interactions. We fully recognize the limitations of this indicator and have
now included a discussion of these concerns in the Discussion section of the
manuscript.

Author response table 2. Adjusted PS-IPW relative risk of malaria, by mother’s
HIV statusType of Vaccine*Adjusted RR (95%CI)**Plasmodium spp.* Parasitemia (Microscopic
Examination of Blood Film)*Plasmodium falciparum* Antigenemia (Rapid
Detection of HRP2)Not adjusting for mother’s HIV statusAdjusting for mother’s HIV statusNot adjusting for mother’s HIV statusAdjusting for mother’s HIV status*BCG*2.71 (1.36-5.39)2.20 (0.68-7.10)3.23 (1.58-6.61)2.17 (0.66-7.10)*DTP*0.38 (0.01-10.1)0.93 (0.23-3.82)0.07 (0.01-0.48)1.34 (0.51-3.54)*Measles*1.00 (0.18-5.45)0.66 (0.30-1.45)0.73 (0.15-3.61)0.86 (0.43-1.72)*Polio*0.60 (0.22-1.62)1.75 (0.56-5.49)0.81 (0.28-2.31)3.27 (1.17-9.15)*Vitamin A*0.24 (0.18-0.34)0.09 (0.04-0.19)0.23 (0.17-0.31)0.13 (0.06-0.27)

*b) Number of doses of each of the immunizations or vitamin A supplements
received (it is likely that older children had more immunizations and A supplements
than younger)*. *Were immunization schedules and supplementation
programmes similar in the four countries?*

Our study models controlled for age both through the use of propensity scores and in the
actual model. As for immunization and vitamin A schedules, Author response table 3 shows
that they were fairly similar across different countries. The main distribution
mechanism was also similar for vitamin A. We have included this information in the first
paragraph of the Results section of the revised manuscript.

Author response table 3. Overview of Immunization Schedules for each
country^1^Type of Vaccination/SupplementationCountryBurkina FasoMozambiqueRwandaSenegal*BCG*BirthBirthBirthBirth*DTP*8,12,16 weeks* (DTP)6,10,14 weeks (DTPHep)6,10,14 weeks (DTPHibHep)6,10,14 weeks (DTPHibHep)*Measles*9 months9 months9 months9 months*Polio*Birth,8,12,16 weeks*6,10,14 weeksBirth, 6,10,14 weeksBirth, 6,10,14 weeks*Vitamin A*Every 6 months from ages 6-59 months *Main Distribution
Mechanism*:Integrated Child Health campaignEvery 6 months from ages 6-59 months *Main Distribution
Mechanism*:Integrated Child Health campaignEvery 6 months from ages 6-59 months *Main Distribution
Mechanism*:Integrated Child Health campaignEvery 6 months from ages 6-59 months *Main Distribution
Mechanism*:Integrated Child Health campaign

*Guidelines report immunization schedule in months (2,3,4 months). Converted into
weeks for this table.

^1^Adapted from: Unicef and WHO. Immunization Summary: The 2007 Edition.
(published February 2007), Vaccination Information Management System: www.vimsdata.org
(Last accessed September 19, 2014) and personal communication with E. Alden (UNICEF) on
December 8, 2014.

*8) Third paragraph in the Results section: the authors indicate that vitamin A
was more protective the greater the time from the last dose. The last panel
of*
Figure 2
*does not seem to show this, nor does the statement tally with the first
paragraph of the Discussion*.

We apologize for the confusion. The increasingly protective effect occurs with malaria
antigenemia but not malaria parasitemia. To avoid any confusion, this phrase has now
been removed from the Discussion.

*9) Fourth paragraph in the Discussion section: the authors suggest that the
increase in antigenemia associated with BCG vaccination might be due to false
positivity from other parasitic infections; also that false–positivity has
been reported in Paracheck assays. However, given low numbers and wide CIs, a type I
error could be a possibility. Also, since different assays were used*,
*can the authors advise whether the increase in antigenemia was similar in the
subsamples when using either one of the assays?*

*Please note that Chagas disease is not found in these countries. As most
children tested were very young, they are unlikely to have significant exposure and
antibodies to* Schistosomiasis.

We thank the reviewers’ for their observations. We have now included an explicit
statement in the fourth paragraph of the Discussion section that Chagas is not endemic
in the countries we studied. As for *Schistosomiasis*, there is growing
evidence to suggest it may be an overlooked endemic disease among infants and young
children in Africa. We have included a statement and reference to this material in the
same section of the text. Finally, in terms of examining increases in antigenemia
stratified by type of RDT, regrettably, the small sample size particularly in the
BCG-negative, malaria-positive cells (2x2 table) led to problems in model convergence
when stratifying results by diagnostic used.